# Diet Quality and Risk of Parkinson’s Disease: The Rotterdam Study

**DOI:** 10.3390/nu13113970

**Published:** 2021-11-07

**Authors:** Anne J. Strikwerda, Lisanne J. Dommershuijsen, M. Kamran Ikram, Trudy Voortman

**Affiliations:** 1Department of Epidemiology, Erasmus MC University Medical Center, P.O. Box 2040, 3000 CA Rotterdam, The Netherlands; a.strikwerda@erasmusmc.nl (A.J.S.); l.dommershuijsen@erasmusmc.nl (L.J.D.); m.ikram@erasmusmc.nl (M.K.I.); 2Department of Neurology, Erasmus MC University Medical Center, P.O. Box 2040, 3000 CA Rotterdam, The Netherlands; 3Division of Human Nutrition and Health, Wageningen University & Research, P.O. Box 9101, 6700 HB Wageningen, The Netherlands

**Keywords:** diet quality, Parkinson’s disease, cohort study, risk factors, etiology

## Abstract

The Mediterranean diet has been associated with the risk of Parkinson’s disease (PD), but limited research has been performed on other dietary patterns. We studied the relationship between overall diet quality and PD risk in the general population. We included 9414 participants from the Rotterdam Study, a prospective population-based study in the Netherlands. Diet was defined using a Dutch diet quality score, a Mediterranean diet score and data-driven dietary patterns constructed with principal component analysis (PCA). During an average follow-up of 14.1 years, PD was diagnosed in 129 participants. We identified a ‘Prudent’, ‘Unhealthy’ and ‘Traditional Dutch’ pattern from the PCA. We found a possible association between the Mediterranean diet (Hazard ratio (HR) per standard deviation (SD) 0.89 (95% confidence interval (CI) 0.74–1.07)), the ‘Prudent’ pattern (HR per SD 0.81 (95% CI 0.61–1.08)) and the risk of PD. However, no associations with PD risk were found for the Dutch diet quality score (HR per SD 0.93 (95% CI 0.77–1.12)), the ‘Unhealthy’ pattern (HR per SD 1.05 (95% CI 0.85–1.29)) or the ‘Traditional Dutch’ pattern (HR per SD 0.90 (95% CI 0.69–1.17)). In conclusion, our results corroborate previous findings of a possible protective effect of the Mediterranean diet. Further research is warranted to study the effect of other dietary patterns on PD risk.

## 1. Introduction

Parkinson’s disease (PD) is estimated to affect over 6 million individuals worldwide; the disease burden is expected to increase even further [1]. The cause of PD is multifactorial, as genetic and environmental factors play an important role [2,3]. Previously, several lifestyle factors had been associated with PD risk [4]. Given the hypothesis that the origin of PD might be found in the gut, and the important influence of diet on the gut microbiome, modification of diet could potentially affect PD risk [5,6,7,8].

Indeed, several studies have linked intake of specific foods or nutrients to PD risk. Frequent intake of certain dairy products has been be associated with an increased risk of PD, whereas higher coffee and tea intake has been associated with a decreased risk of PD [9,10,11,12,13,14]. For other foods, inconsistent results have been described [15,16,17,18]. In addition to food groups, complete dietary patterns have been identified as a comprehensive way to study the role of diet in chronic diseases [19]. Several studies have reported a reduced risk of PD with better adherence to the Mediterranean diet, but inconsistent associations were found between the American Healthy Diet Index and PD risk [20,21,22,23]. The question thus remains whether it is specifically the Mediterranean diet or a ‘healthy’ dietary pattern in general that is associated with PD risk.

In this study, we determined the association between diet and the risk of PD in the general population, through pre-defined diet scores and data-driven dietary patterns. Additionally, we determined the effects of specific food groups on the risk of PD to further unravel the association between diet and PD.

## 2. Materials and Methods

### 2.1. Study Population

The Rotterdam Study is a prospective population-based cohort study that aims to examine etiology, natural history and targets for intervention in chronic diseases. The Rotterdam Study was initiated in 1990, recruiting all inhabitants aged 55 years or older from a defined suburban district in Rotterdam (cohort Rotterdam Study-I, RS-I). The cohort was extended in 2000 (recruiting participants ≥ 55 years) and again in 2006 (recruiting participants ≥ 45 years), forming cohort RS-II and RS-III, respectively. At baseline, and at every three to five year follow-up visit, participants took part in a home interview and a wide set of examinations at the research center [24].

The baseline cohort of the Rotterdam Study consisted of 14,926 participants (RS-I: 7983, RS-II: 3011, RS-III: 3932). From this group, dietary data were available for 9788 participants. Based on previous studies, we determined a reported energy intake of less than 500 kcal/day or more than 5000 kcal/day as unreliable and, therefore, excluded 88 participants [25]. From the remaining 9700 participants with reliable and available dietary data, participants with prevalent parkinsonism or PD (*n =* 51), without a screening for parkinsonism at baseline (*n =* 174) or who did not give consent for parkinsonism follow-up (*n =* 61), were excluded. This resulted in a final eligible cohort of 9414 participants for the current study. The flowchart of study participants can be found in Figure 1.

### 2.2. Dietary Intake Assessment and Dietary Pattern Construction

Diet was assessed at baseline with semi-quantitative food frequency questionnaires (FFQ), which were validated for the ranking of food and nutrient intake in the Rotterdam Study and other populations [26,27]. The FFQ evaluated the intake of 170 food items in the preceding year for RS-I and RS-II and was upgraded to evaluate 389 food items in the preceding month for RS-III. Energy and nutrient intake were calculated using Dutch food composition tables (NEVO) [28].

### 2.3. Dietary Patterns

Adherence to the Dutch dietary guidelines was determined through a Dutch diet quality score, which was previously described in more detail [25]. In short, the score is based on adherence to fourteen components of the 2015 Dutch food-based dietary guidelines [29]. A score of zero was given to participants who did not meet the criterion and a score of one for participants who met the criterion, resulting in a maximum sum score of fourteen points. Adherence to the Mediterranean diet was determined with the Mediterranean diet score that was created by Panagiotakos et al. [23]. This sum score is based on eleven main components of the Mediterranean diet. Participants were assigned zero to five points for adherence to each component, resulting in a maximum sum score of 55 points [23].

In addition to pre-defined diet scores, we created dietary patterns derived from actual food intake in the population in order to determine differences in effects as compared to the pre-defined diet scores. For this, food items from the FFQ were combined into 23 food groups based on nutrient composition and everyday use. The food groups are described in Appendix A. These food groups were used for the construction of dietary patterns with principal component analysis (PCA) and Varimax rotation for better interpretation. Based on scree plots, we included components with an eigenvalue above 1.5 for further analyses with PD risk. A component is defined by a cluster of food groups, where each food group has a factor loading that indicates the strength of the association between the food group and the component. To characterize dietary patterns from the components, we focused on factor loadings below −0.3 or higher than 0.3. For each participant and for each of the dietary patterns standardized adherence scores (Z-scores) were calculated by summing up the intake per food group and weighing this by the factor loading corresponding to that food group. To determine the effects of the individual food groups on PD risk, all 23 food groups constructed for the PCA were also analyzed separately.

### 2.4. Parkinson’s Disease Ascertainment

Incident PD during follow-up was ascertained through continuous linkage with clinical records from general practitioners, nursing home physicians, hospitals and pharmaceutical records. In addition, participants could self-report a new diagnosis of PD during the home interview and were screened for parkinsonian symptoms during follow-up visits. All case reports of incident events were evaluated by a panel led by an experienced neurologist, where the final diagnosis was confirmed. This process has previously been described more thoroughly [30]. Incident parkinsonism included: PD, drug-induced parkinsonism, vascular parkinsonism, Lewy body disease, parkinsonism with dementia other than Lewy body disease, multi-system atrophy, progressive supranuclear palsy, corticobasal degeneration or parkinsonism resulting from a tumor. Follow-up continued until the first of January 2018, or until ascertained parkinsonism, death, or loss to follow-up, whichever came first.

### 2.5. Covariate Assessment

Covariate data were collected from the study round in which the FFQ was completed. Weight and height were measured at the research center, from which body mass index (BMI) was calculated by dividing weight by height squared (kg/m²). Further covariate data were self-reported using questionnaires. Smoking behavior was determined as all forms of tobacco smoking and was classified into never, former or current smoking. Levels of education were low (primary education), lower-middle (lower or intermediate general or lower vocational education), middle (intermediate vocational or higher general education) or high (higher vocational education or university). Genetic ancestry based on genome-wide association studies (GWAS), was estimated using Admixture [31]. Ancestral groups were created including individuals having at least 50% genetic material from that group, and individuals with less than 50% genetic material in any group were classified as ‘Admixed’. The groups of genetic ancestry were European, East Asian, African and Admixed.

### 2.6. Statistical Analysis

Missing values of genetic ancestry (10.2%), education (0.6%), BMI (0.7%) and smoking behavior (0.5%) were multiply imputed using all covariates, the Dutch diet quality score and the outcome as predictors (5 imputations). We used Cox proportional hazards models to assess the association between the pre-defined diet scores, the data-driven dietary patterns, the food groups and PD risk. Hazard ratios (HR) and 95% confidence intervals (CI) were computed and calculated per standard deviation (SD) increase for the diet scores, the dietary patterns and the food groups. Additionally, we calculated the HRs for the diet scores and dietary patterns per tertile, with the lowest tertile as reference. The basic model was adjusted for sex, age and cohort and the covariate model was additionally adjusted for education, smoking behavior, BMI and energy intake. The proportional hazard assumption was checked with Schoenfeld residuals; there were no clear violations [32].

We performed several sensitivity analyses to examine the robustness of our results. First, we reran the analysis for different episodes of follow-up to compare short- versus long-term effects. Second, we studied the association between the dietary patterns and PD for the cohorts RS-I, RS-II and RS-III separately to assess whether the difference in FFQ or other diet-related cohort differences affected our results. Finally, we evaluated differences in effects between males and females and evaluated whether using all-cause parkinsonism as the outcome changed our results.

Instead of dichotomizing the found effects into ‘significant’ and ‘non-significant’, we will show effect estimates and their corresponding confidence intervals in order to describe the range of estimates compatible with the data. All analyses were performed in R.4.0.3 (R Core Team, Vienna, Austria) [33].

## 3. Results

### 3.1. Population Characteristics

The characteristics of the study population are shown in Table 1. The median age of the study population was 62.2 years (interquartile range, IQR (58–70)) and 57.8% were female. Most participants were current or former smokers (67.8%) and had a low or lower-middle level of education (56.6%). The median energy intake was 2020 kcal/day (IQR (1682–2410)), the median Dutch diet quality score was 7.0 out of 14 (IQR (5–8)) and the median Mediterranean diet score was 37 out of 55 (IQR (34–39)). After a mean follow-up of 14.1 years, 129 participants were diagnosed with PD. The mean age at diagnosis of PD was 76.6 years (standard deviation, SD ± 7.4) and 47.3% of the participants with incident PD were female.

### 3.2. Dietary Patterns and Parkinson’s Disease Risk

Using PCA, we identified three components with an eigenvalue above 1.5 for which the Varimax rotated factor loadings are presented in Table 2. We named the dietary patterns based on the factor loadings of the clustered food groups. The pattern we named ‘Prudent’ was defined by a high intake of vegetables, legumes, white meat, fish and nuts. The ‘Unhealthy’ pattern was defined by high intake of alcoholic beverages, coffee, eggs, red or processed meat, animal-based fats and by low intake of fruits and tea. The ‘Traditional Dutch’ pattern was defined by high intake of cheese, vegetable oils and spreads, potatoes and fries, wholegrain products and sweet snacks. The three patterns explained 8.6%, 7.5% and 7.7% of the total variance, respectively, resulting in a total explained variance of 23.8%.

Associations between the diet scores, data-driven dietary patterns and the risk of PD are shown in Table 3. No associations with the risk of PD were observed for the Dutch diet quality score, the ‘Unhealthy’ pattern and the ‘Traditional Dutch’ pattern, neither in the continuous nor the categorical analysis (adjusted HRs per SD: Dutch diet quality score 0.93 (95% CI 0.77–1.12), ‘Unhealthy’ pattern 1.05 (95% CI 0.85–1.29), ‘Traditional Dutch’ pattern 0.90 (95% CI 0.69–1.17)). Estimates of the unadjusted and adjusted models for the Mediterranean diet score (adjusted HR per SD 0.89 (95% CI 0.74–1.07)) and the ‘Prudent’ pattern (adjusted HR per SD 0.81 (95% CI 0.61–1.08)) suggested a lower hazard of PD with higher scores, although confidence intervals were wide.

Figure 2 shows the associations between the individual food groups from the PCA and the risk of PD. A higher intake of yoghurt and fermented milk (adjusted HR per SD 1.13 (95% CI 0.98–1.31)), animal-based fats (adjusted HR per SD 1.19 (95% CI 1.00–1.43)) and sweet snacks (adjusted HR per SD 1.21 (95% CI 1.03–1.41)) was associated with a higher risk of PD. Although confidence intervals were wide, we observed a lower risk of PD with a higher intake of fruits (adjusted HR per SD 0.82 (95% CI 0.63–1.07)), vegetables (adjusted HR per SD 0.84 (95% CI 0.65–1.09)), cheese (adjusted HR per SD 0.79 (95% CI 0.62–1.01)) and savory snacks (adjusted HR per SD 0.73 (95% CI 0.45–1.19)).

### 3.3. Sensitivity Analyses

We observed no clear difference in associations between the dietary patterns and the risk of PD with different follow-up times or for the three cohorts separately (Appendix A). When stratifying for sex, stronger effect estimates were found in the association of dietary patterns with the risk of PD for males, although confidence intervals overlapped between the two strata (Appendix A). Finally, changing the outcome to all-cause parkinsonism increased the number of total incident events in the eligible cohort to 254. In this analysis, the association between the ‘Prudent’ dietary pattern and the Mediterranean diet score with risk of all-cause parkinsonism became more pronounced (Appendix A: ‘Prudent’ pattern adjusted HR per SD 0.76 (95% CI 0.61–0.95), Mediterranean diet adjusted HR per SD 0.86 (95% CI 0.76–0.98)).

## 4. Discussion

In this population-based cohort study, we corroborated previous findings of a relation between the Mediterranean diet and the risk of PD. However, we did not find clear effects of adherence to the Dutch dietary guidelines on the risk of PD. In addition, our study provides new evidence for the possible protective and detrimental effects of separate food groups on the risk of PD.

A previous Finnish cohort study found no significant association between the Alternate Healthy Eating Index (AHEI)—a diet score reflecting US guidelines—and risk of PD, while an American cohort study found a lower risk of PD with a higher AHEI score [20,22,34]. However, in the same cohort studies, and also more recently in a Swedish cohort study, a higher Mediterranean diet score was consistently associated with a lower risk of PD [20,21,22,23,35]. These previous studies are in line with our findings of a possible association between the Mediterranean diet and the ‘Prudent’ dietary pattern and the risk of PD, without clear effects of the Dutch diet quality score.

Our more detailed food group analyses revealed that higher intake of yoghurt and fermented milk, animal-based fats and sweet snacks was associated with a higher risk of PD. The association between intake of sweet snacks and PD is novel; it is yet unclear what underlies this association. Previous research on impulsive behavior suggested higher impulsivity in PD patients, and possibly, the impulsive intake of sweet snacks could explain the found association [36]. This might also imply that snacking is a consequence of prodromal PD rather than a cause, which needs further investigation. Based on previous research, we also hypothesized that a frequent intake of dairy products, mostly milk intake, would be associated with an increased PD risk, but we found inconsistent results concerning this association [12,14]. We found no association between higher milk and cream intake and PD, but we did find a possible higher risk of PD for yoghurt and fermented milk intake and animal-based fat intake, which mostly consists of dairy butter. The positive association between dietary animal fat intake and PD is in line with results from a previous case–control study by Logroscino et al. [15]. However, the association remains inconsistent in the literature, as Chen et al. found no significant association between animal fat intake and PD in the Nurses’ Health Study and the Health Professionals Follow-up Study [16]. The possible associations we found between cheese and savory snacks and the risk of PD are remarkable and have not been reported by previous studies, and thus, require further study.

Our study has several strengths. First, the population-based study design of the Rotterdam Study with a continuous follow-up of participants through clinical records and follow-up visits reinforces the validity of the incident PD events in this study. Second, by using pre-defined diet scores, data-driven dietary patterns and separate food groups, we were able to secure a complete image of diet quality. In addition, with both pre-defined diet scores and data-driven dietary patterns derived from the PCA, we managed to explore and compare PD risk based on both desired intake reflecting guidelines and the actual diet patterns from the data. Using this two-faced approach, we observed similar results for the Mediterranean diet score and the comparable ‘Prudent’ data-driven dietary pattern, which strengthens our conclusions. We found only one study that has used the combination of data-driven patterns and pre-defined diet scores in association with PD [22].

Nonetheless, there are also limitations in our study. Firstly, we used only baseline diet information. As dietary habits are subject to changes over time, diet could have altered during the follow-up period, which was not taken into account in the current study [37]. Secondly, the low number of incident PD events in our study gave rise to wide confidence intervals around our estimates, which indeed became remarkably smaller when more events were included by using all-cause parkinsonism as the outcome. Thirdly, our study sample included solely Dutch individuals living in the Rotterdam district Ommoord; the dietary patterns identified in this study and their associations with PD are thus not necessarily generalizable to other study populations. Lastly, two different FFQs were used at baseline to assess dietary intake, which could have resulted in differences in the determinant between the study cohorts. Nevertheless, both FFQs included the necessary items to calculate data-driven dietary patterns and the specific food groups. Moreover, repeating the analyses for each cohort separately revealed similar results, implying that this difference in questionnaires did not impact the findings much.

Taking into account both our findings and the findings of previous studies, the Mediterranean diet seems to be a promising focus for future research aiming to study the preventive potential of lifestyle interventions for PD [20,22,35]. Nevertheless, the mechanisms underlying the association between the Mediterranean diet and PD are still to be resolved. Potentially, its anti-inflammatory character and ability to reduce oxidative stress could contribute to the beneficial effect on PD risk [7,38]. These mechanisms, as well as potential beneficial effects of other healthy dietary patterns, remain subject to further investigation.

## 5. Conclusions

In summary, our results corroborate previous findings of an association between the Mediterranean diet and the risk of PD. In addition, our results suggest a possible link between dairy products, fruits, vegetables and sweet and savory snacks and the risk of PD, which warrants replication by future studies.

## Figures and Tables

**Figure 1 nutrients-13-03970-f001:**
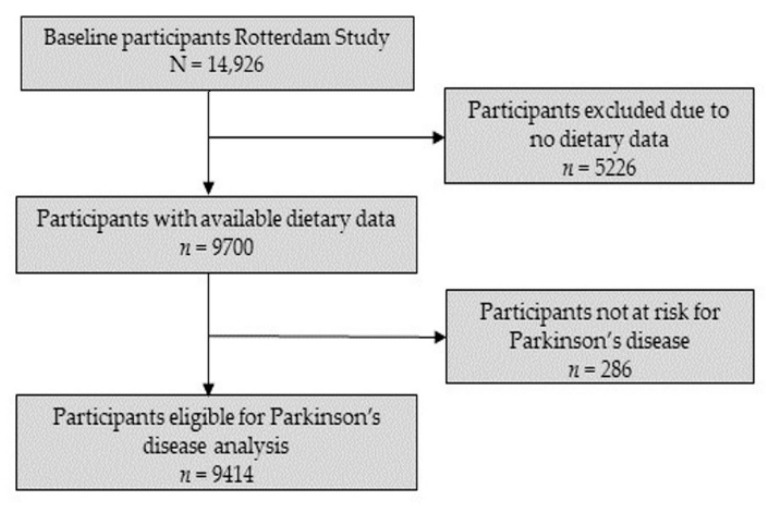
Flowchart for selection of study participants.

**Figure 2 nutrients-13-03970-f002:**
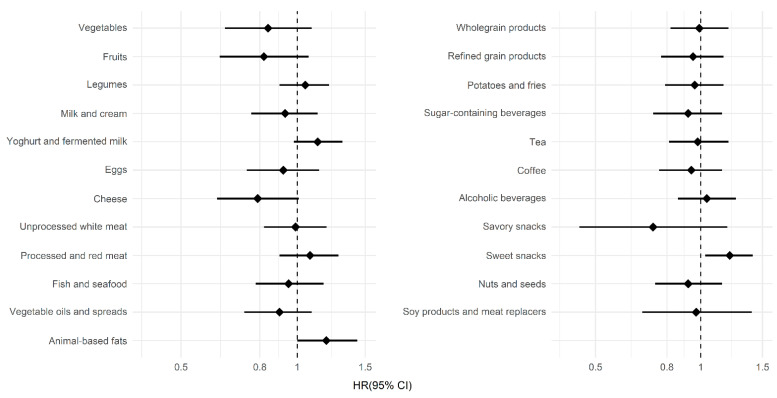
Associations between the food groups and Parkinson’s disease. The hazard ratios (HR), shown as diamonds, and the 95% confidence interval (CI), shown as lines, were obtained using Cox proportional hazard models. Hazard ratios are shown per standard deviation increase in intake for the 23 food groups, which were also used in the principal component analysis.

**Table 1 nutrients-13-03970-t001:** Baseline characteristics.

	*n =* 9414
Sex, female	5439 (57.8%)
Age, years	62.2 (58.0–70.0)
Genetic ancestry, European ^a^	9251 (98.3%)
Education ^a^	
Low	1456 (15.5%)
Lower-middle	3872 (41.1%)
Middle	2631 (27.9%)
High	1455 (15.5%)
Smoking behavior ^a^	
Never	3023 (32.1%)
Former	4178 (44.4%)
Current	2213 (23.5%)
Alcohol intake, drinks/day	0.5 (0.0–1.8)
BMI, kg/m² ^a^	26.8 (±4.0)
Energy intake, kcal/day	2020 (1,682–2,410)
Dutch diet quality score ^b^	7.0 (5.0–8.0)
Mediterranean diet score ^c^	37.0 (34.0–39.0)

Values are median (interquartile range, IQR), mean (standard deviation, ±SD) or n (%). ^a^ Genetic ancestry, education, smoking behavior and body mass index (BMI) are based on imputed values. The number of missing values were 969 for genetic ancestry, 55 for education, 45 for smoking behavior and 70 for BMI. ^b^ Theoretical range 0–14, where a higher score is better adherence to the Dutch dietary guidelines. ^c^ Theoretical range 0–55, where a higher score is better adherence to the Mediterranean diet.

**Table 2 nutrients-13-03970-t002:** Factor loadings for each group per dietary pattern.

Food Groups	Loadings Per Dietary Pattern		
	Prudent	Unhealthy	Traditional Dutch
Vegetables	**0.56**	−0.23	0.15
Fruits	**0.41**	**−0.44**	-
Legumes	**0.46**	-	-
Milk and cream	−0.11	-	0.18
Yoghurt and fermented milk	-	**−0.38**	-
Eggs	0.28	**0.32**	0.12
Cheese	-	-	**0.39**
Unprocessed white meat	**0.41**	-	-
Processed and red meat	-	**0.43**	**0.44**
Fish and seafood	**0.51**	-	−0.13
Vegetable oils and spreads	-	−0.16	**0.63**
Animal-based fats	-	**0.37**	0.12
Wholegrain products	0.14	**−0.32**	**0.54**
Refined grain products	**0.35**	0.25	-
Potatoes and fries	−0.17	-	0.62
Sugar-containing beverages	0.27	0.24	-
Tea	-	**−0.47**	-
Coffee	-	**0.34**	**0.34**
Alcoholic beverages	0.25	**0.46**	-
Savory snacks	**0.37**	0.21	-
Sweet snacks	-	-	0.32
Nuts and seeds	**0.45**	-	-
Soy products and meat replacers	**0.32**	−0.18	-
% of explained variance	8.6	7.5	7.7

Dietary patterns and factor loadings were obtained from the principal component analysis. The three components with an eigenvalue >1.5 are presented in the table as the ‘Prudent’, ‘Unhealthy’ and ‘Traditional Dutch’ dietary pattern, which were named based on the factor loadings belonging to the clustered food groups in each component. The presented loadings were rotated with Varimax rotation for better interpretation. All factor loadings that are higher than 0.3 and lower than −0.3 are printed in bold. Details on the food groups are presented in Appendix A.

**Table 3 nutrients-13-03970-t003:** Associations between dietary patterns and Parkinson’s disease (*n*=9414).

Dietary Pattern	No. of Incident Parkinson’s Disease	Basic Model HR (95% CI)	Covariate Model HR (95% CI)
Dutch diet quality score per SD	129	0.97 (0.81–1.17)	0.93 (0.77–1.12)
Tertiles			
Low (reference)	37	1	1
Medium	48	1.02 (0.66–1.59)	0.97 (0.62–1.52)
High	44	1.07 (0.68–1.69)	0.98 (0.62–1.55)
Mediterranean diet score per SD	129	0.93 (0.78–1.12)	0.89 (0.74–1.07)
Tertiles			
Low	38	1	1
Medium	57	1.39 (0.92–2.11)	1.33 (0.87–2.02)
High	34	0.88 (0.55–1.40)	0.80 (0.50–1.29)
Prudent pattern per SD	129	0.87 (0.67–1.14)	0.81 (0.61–1.08)
Tertiles			
Low (reference)	53	1	1
Medium	47	0.97 (0.65–1.46)	0.94 (0.63–1.42)
High	29	0.95 (0.57–1.56)	0.89 (0.52–1.50)
Unhealthy pattern per SD	129	0.98 (0.80–1.21)	1.05 (0.85–1.29)
Tertiles			
Low (reference)	44	1	1
Medium	48	1.07 (0.70–1.62)	1.15 (0.75–1.74)
High	37	0.96 (0.60–1.52)	1.08 (0.67–1.74)
Traditional Dutch pattern per SD	129	0.99 (0.82–1.20)	0.90 (0.69–1.17)
Tertiles			
Low (reference)	28	1	1
Medium	51	1.50 (0.94–2.41)	1.44 (0.88–2.35)
High	50	1.27 (0.77–2.10)	1.16 (0.64–2.10)

The hazard ratios (HR) and 95% confidence interval (CI), obtained using Cox proportional hazard models, are shown per standard deviation (SD) increase for the dietary patterns and scores. The basic model was adjusted for sex, age at baseline and Rotterdam Study cohort. The covariate model was adjusted for all items in the basic model and additionally for body mass index (BMI), education, smoking behavior and energy intake.

## Data Availability

Data can be obtained upon request. Requests should be directed towards the management team of the Rotterdam Study (datamanagement.ergo@erasmusmc.nl), which has a protocol for approving data requests. Because of restrictions based on privacy regulations and informed consent of the participants, data cannot be made freely available in a public repository.

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
