# Peer review of "Diet Quality and Risk of Parkinson’s Disease: The Rotterdam Study"

_nutrients, 2021, doi:10.3390/nu13113970_

Round 1
Reviewer 1 Report
In the introduction, the authors refer to milk intake as one of the possible riks in the development of PD. This citation is appropriate, but there are a number of articles that point the opposite. For example Olsson et al. (2020) showed a relationship between fermented milk and PD increased risk
( Olsson E., J., Milk and Fermented Milk Intake and Parkinson's Disease: Cohort Study. Nutrients: 2020 Sep 10;12(9):2763. doi: 10.3390/nu12092763).
So, it would be more useful to report these and state how tha data are conflicting and therefore all to be evaluated.
Reviewer 2 Report
-there is no report of ethnic roots of the partecipants. It is conceivable that people with mediterranean diet may come from mediterranean areas and therefore have different genetics background. The authors should specify the ethnic origin of the re three diet groups as depicted by PCA since no genetics is reported.
- PCA is quite a complicated way of saying simple things: it would be advisable to run also more simple analysis/statistics and explain results in a more understandable way
Reviewer 3 Report
This is a well written manuscript by Strikwerda et al. that analyzes the associate between diet and HR for Parkinson’s Disease. The manuscript is direct and concise. The work is interesting and appears to convey both confirmatory and new results, both of which should be valued.
Minor revisions:
In : “Given the presumable origin of PD pathology”; can this be cited? It seems that PD has a strong association with the gut microbiome; it could be causative, may not be the sole cause. It could be a marker for dysregulation elsewhere. Seems like saying this a little differently would be good.
Question: how are the cutoffs for reliable data determined, i.e. why are they 500 and 5000 kcal?
How is smoking (only cigarettes? or others) determined?
Was there any association of smoking with HR?
Is there a possibility to discuss relationship of diets to metabolism (gut bacterial metabolism and/or serum/organ metabolism?) and therefore underlying mechanisms of PD?
Race and Ethnicity can be considered, although cautiously {Flanagin, et al. JAMA 2021, 326, 621-627, DOI 10.1001/jama.2021.13304}. Was this considered?
Style/Writing:
Check with journal, but citations often appear after the period.
Figure 1 : flow chart is a bit redundant but is appreciated – could be made smaller though.
Figure 2 caption appears under-developed; although it is noted in the figure, would be good for caption to explain that hazard ratios (diamonds) are displayed with 95% confidence intervals (lines); would like to see it be more descriptive and highlight key trends that are discussed in the main text. Remember, most readers skim figures/captions before deciding to read the paper so I advise to make the Fig 2 caption richer.
I would say this also for Table 2 caption; could be more developed.
In “A score of zero was given to participants who did not meet the criterion and a score of 82 one for participants who met the criterion, resulting in a maximum sum-score of fourteen points.” this approach is termed a “l0 norm” in other disciplines. Not sure if that is common terminology in this community, but just wanted to check. In other words, this ‘sum-score’ could be called an ‘l0 norm’. On the other hand, the Mediterranean sum-score is actually a “l1-norm”. But if these terms are not commonly done in this discipline then there is no reason to do so here.
In statistical analysis section, good to state software used.
Better to say “The mean age at diagnosis of PD was...”
Round 2
Reviewer 2 Report
The paper has not been substantially improved.
The authors should reply point by point to the comments: this has not been done or provided.
The authors should embody in the discussion the core of my questions and their replies above all about genetics and the different geographic areas of the subjects of the study.
Author Response
We would like to thank the reviewers for their comments and suggestions on our manuscript. We have provided our response to the reviewers’ specific comments below. Our previously formulated comments were accidentally not uploaded in the text box. We apologize for this and have now uploaded our response both in a PDF file and in the text box and included another adjustment for the comment concerning ethnicity. Citations of the adapted text in the manuscript are provided whenever appropriate and are shown in indented text. The page numbers refer to the updated manuscript file.
Comment: There is no report of ethnic roots of the partecipants. It is conceivable that people with mediterranean diet may come from mediterranean areas and therefore have different genetics background. The authors should specify the ethnic origin of the re three diet groups as depicted by PCA since no genetics is reported.
Response: We agree with the reviewer that differences in genetic background may explain differences observed in diet-disease associations between different studies. We have now reported the genetic ancestry of the participants in Table 1: Baseline characteristics, please find the adapted table below. Additionally we have now written about this matter in the limitations in our discussion. Because 98.3% of our study population was of European ancestry, and other groups very small, we could not pick up any differences between groups. Unfortunately, ‘European’ ancestry does not distinguish Northern European from Southern European. Nevertheless, all participants were recruited in Ommoord, a neighbourhood in Rotterdam, where mostly Dutch people live. Further studies in multiethnic populations are needed to further explore potential differences.
“Genetic ancestry based on genome-wide association studies (GWAS), was estimated using Admixture [31]. Ancestral groups were created by individuals having at least 50% genetic material from that group and individuals with less than 50% genetic material in any group were classified as ‘Admixed’. The groups of genetic ancestry were European, East-Asian, African and Admixed.” (Page 3 and 4, Materials and Methods)
“Thirdly, our study sample included solely Dutch individuals living in the Rotterdam district Ommoord, the dietary patterns identified in this study and their associations with PD are thus not necessarily generalizable to other study populations.” (Page 9, Discussion)
Comment: PCA is quite a complicated way of saying simple things: it would be advisable to run also more simple analysis/statistics and explain results in a more understandable way
Response: In this article, our main analysis was based on a simple Dutch diet score sum score and Mediterranean diet score sum score. As a sub-analysis, we performed the more complex PCA to reduce our multi-level data to dietary patterns that represent actual intake in the population in order to determine if we found differences as compared to the pre-defined diet scores. Because of the complexity of this analysis, we have further clarified the rationale, methods and results of this analysis.
” In addition to pre-defined diet scores, we created dietary patterns derived from actual food intake in the population in order to determine if we found differences as compared to the pre-defined diet scores.” (Page 3, Materials and Methods)
“Based on screeplots, we included components with an eigenvalue above 1.5 for further analyses with PD risk. A component is defined by a cluster of food groups, where each food group has a factor loading that indicates the strength of the association between food group and the component. (Page 3, Materials and Methods)
“Using PCA, we identified three components with an eigenvalue above 1.5 for which the Varimax rotated factor loadings are presented in Table 2. Based on the factor loadings of the clustered food groups we named the dietary patterns.” (Page 5, Results)
“Using this two-faced approach we observed similar results for the Mediterranean diet score and the comparable ‘Prudent’ data-driven dietary pattern, which strengthens our conclusions.” (Page 9, Discussion)
Table 1. Baseline characteristics
|
N = 9,414 |
|
|
Sex, female |
5,439 (57.8%) |
|
Age, years |
62.2 [58.0-70.0] |
|
Genetic ancestry, Europeana |
9,251 (98,3%) |
|
Educationa |
|
|
Low |
1,456 (15.5%) |
|
Lower-middle |
3,872 (41.1%) |
|
Middle |
2,631 (27.9%) |
|
High |
1,455 (15.5%) |
|
Smoking behaviora |
|
|
Never |
3,023 (32.1%) |
|
Former |
4,178 (44.4%) |
|
Current |
2,213 (23.5%) |
|
Alcohol intake, drinks/day |
0.5 [0.0-1.8] |
|
BMI, kg/m²a |
26.8 (±4.0) |
|
Energy intake, kcal/day |
2,020 [1,682-2,410] |
|
Dutch diet scoreb |
7.0 [5.0-8.0] |
|
Mediterranean diet scorec |
37.0 [34.0-39.0] |
